

# Evaluation of P5CS and ProDH activity in *Paulownia tomentosa* (Steud.) as an indicator of oxidative changes induced by drought stress

Joanna Kijowska-Oberc[1], Mikołaj K. Wawrzyniak[1], Liliana Ciszewska[2] and Ewelina Ratajczak[1]

[1] Institute of Dendrology, Polish Academy of Sciences, Kórnik, Wielkopolskie, Polska
[2] Laboratory of RNA Biochemistry, Institute of Molecular Biology and Biotechnology, Faculty of Biology, Adam Mickiewicz University in Poznan, Poznan, Wielkopolskie, Polska

## ABSTRACT

The aim of the study was to investigate changes in proline metabolism in seedlings of tree species during drought stress. One month old *Paulownia tomentosa* seedlings were exposed to moisture conditions at various levels (irrigation at 100, 75, 50 and 25% of field capacity), and then the material (leaves and roots) was collected three times at 10-day intervals. The activity of enzymes involved in proline metabolism was closely related to drought severity; however, proline content was not directly impacted. The activity of pyrroline-5-carboxylate synthetase (P5CS), which catalyzes proline biosynthesis, increased in response to hydrogen peroxide accumulation, which was correlated with soil moisture. In contrast, the activity of proline dehydrogenase (ProDH), which catalyzes proline catabolism, decreased. Compared to proline, the activity of these enzymes may be a more reliable biochemical marker of stress-induced oxidative changes. The content of proline is dependent on numerous additional factors, *i.e.*, its degradation is an important alternative energy source. Moreover, we noted tissue-specific differences in this species, in which roots appeared to be proline biosynthesis sites and leaves appeared to be proline catabolism sites. Further research is needed to examine a broader view of proline metabolism as a cycle regulated by multiple mechanisms and differences between species.

## INTRODUCTION

Drought stress is among the main consequences of climate change (*Hasanuzzaman et al., 2013*). Changes in environmental conditions, such as decreased precipitation, shape the geographical ranges of many tree species (*Dyderski & Jagodziński, 2018*; *Kahveci, Alan & Köse, 2018*) by increasing old-growth forest mortality, reducing species dispersion and leading to the acquisition of new habitats (*Fang et al., 2017*). Water availability affects tree metabolism, which further results in the reproduction and development of offspring (*Dobrowolska, 2015*; *Bogdziewicz et al., 2020*). Drought is especially threatening to trees due to their longevity. As a serious environmental stress factor, drought could cause species to

Corresponding author
Joanna Kijowska-Oberc,
joberc@man.poznan.pl

adapt to new conditions by metabolism modulations; however, drought could also cause organisms to perish (*Anderegg et al., 2015*; *Solarik et al., 2018*).

In response to unfavorable conditions, such as drought, plants accumulate proline (*Szabados & Savoure, 2010*). Proline is among the osmolytes—solutes that accumulate at high concentrations in cells and prevent loss of proper turgor during water stress (*Sigala et al., 2020*). Because it is a component of many membrane proteins, proline stabilizes the integrity of cell membranes, which enables physiological processes to correctly function (*Schertl et al., 2014*). Moreover, proline improves the viability of seedlings when water availability is limited during their initial growth (*Liang et al., 2013*; *Kaur & Asthir, 2015*). Proline is an important component of the nonenzymatic part of the antioxidant system. In addition, proline prevents oxidative damage caused by reactive oxygen species (ROS), including hydrogen peroxide ($H_2O_2$), that accumulate excessively in response to stress (*Signorelli et al., 2014*). It acts as an ROS scavenger and thus retains redox homeostasis in cells (*Kaul, Sharma & Mehta, 2008*; *Kaur et al., 2011*; *Liang et al., 2013*). However, proline content in tissue depends on the levels of its biosynthesis and catabolism (*Székely et al., 2008*).

Housekeeping biosynthesis of proline occurs in the cytoplasm (Fig. 1). Under water stress conditions, proline is synthesized mainly from glutamate, but depending on conditions or plant species, it can be synthetized from ornithine. The first stage in the glutamate pathway is catalyzed by P5C synthetase (P5CS), while NADPH is an electron donor. Glutamate is reduced to glutamyl-5-semialdehyde (GSA), which is spontaneously converted to pyrroline-5-carboxylate (P5C). The activity of P5CS, which controls stress-dependent proline biosynthesis, is upregulated by water and osmotic stress and stimulated by $H_2O_2$ (*Savouré et al., 1995*; *Alvarez, Savouré & Szabados, 2022*). Degradation of proline occurs in mitochondria and is catalyzed by proline dehydrogenase (ProDH), which has been found to be located in mitochondrial membranes (*Cabassa-Hourton et al., 2016*). Proline is oxidized to P5C by ProDH, while FAD+ is reduced to $FADH_2$. The catabolic activity starts when an excessive amount of proline is present in plant cells (*Verslues & Sharma, 2010*). ProDH activity is downregulated by water stress and upregulated by accumulating proline (*Kiyosue et al., 1996*; *Misra & Gupta, 2005*). Therefore, environmental and endogenous signals appear to tightly regulate proline metabolism *via* ROS (*Ben Rejeb, Abdelly & Savouré, 2014*). On the other hand, enhanced proline synthesis under stress maintains the redox potential at values suitable for proper plant development (*Hare, Cress & van Staden, 1999*; *Hare, Cress & van Staden, 2003*). However, although the metabolism mechanism of proline is well known, it has been studied using short-lived plants. Biochemical processes occurring in tree seedlings, which have a much longer development than that of herbaceous plants, may differ from the processes observed in other plants, such as Arabidopsis (*Hare, Cress & Van Staden, 2003*), maize (*Karalija & Selović 2018*) or sweet corn (*Wen et al., 2013*).

*Paulownia tomentosa* (Steud.) is distributed in eastern and central China, but because of its commercial and ornamental value, it is currently cultivated worldwide (*Magar et al., 2016*; *Huber et al., 2023*; *EPPO, 2023*). Due to the desirable properties of timber, such as clear wood grain, high dimensional stability, and resistance to bending, cracking or rotting (*Yaycili & Alikamanoglu, 2005*; *Icka, Damo & Icka, 2016*), it is a very important species for

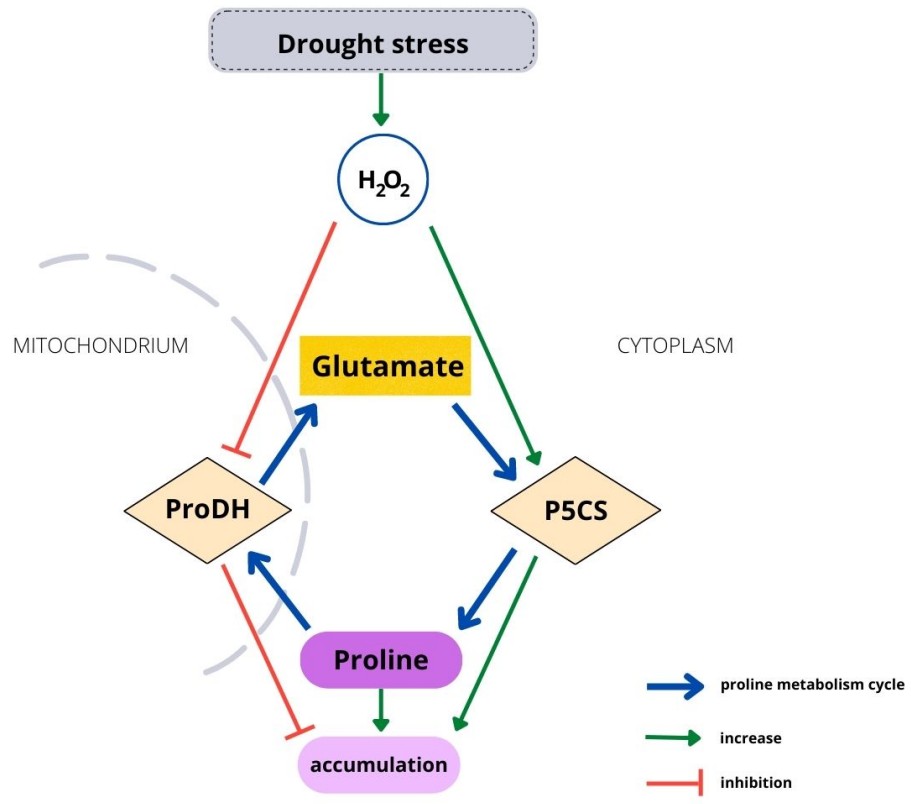

**Figure 1** **Relationship between $H_2O_2$ and the proline metabolism cycle.**

the production of many objects, such as furniture, musical instruments, construction, paper or even bio fuel (*Doumett et al., 2008*; *Akyildiz & Kol, 2010*). *P. tomentosa* also exhibits high decorative qualities; thus, this large arborescent tree is planted in many cities, and its high adaptability has been observed in other countries with a different climate than China (*Essl, 2007*; *Dubova, Voitovych & Boika, 2019*). Due to its high transpiration rate, rapid growth and elevated biomass production, Paulownia has been used as a model woody plant for many studies (*Doumett et al., 2008*; *Ivanova et al., 2014*; *Fan et al., 2016*; *Magar et al., 2018*; *Stefanov et al., 2021*), including deep experiments (*Xu et al., 2014*; *Niu et al., 2016*; *Mohamad, Awad & Gendy, 2021*). Proline plays an important role in the development of woody plants. We have noticed that proline accumulates during drought stress, and the extend by which drought stress affects proline accumulation is higher for species producing more massive and more desiccation-sensitive seeds (recalcitrant category) as well as deciduous species (*Kijowska-Oberc, Dylewski & Ratajczak, 2023*). Moreover, it has been observed that proline content during seed maturation changes and correlates with moisture and thermal conditions only in the case of recalcitrant seeds (*Kijowska-Oberc et al., 2020*).

In this paper, the biochemical response of Paulownia seedlings to different drought stress levels was analyzed based on the levels of accumulation of proline and $H_2O_2$, and the

activity of P5CS and ProDH. The aim was to determine whether changes in the activity of P5CS and ProDH can be an indicator of oxidative changes induced by drought stress.

## MATERIAL AND METHODS

### Experimental design

The seeds were collected from a single tree of *P. tomentosa* growing in the Kórnik Arboretum (western Poland) and were stored under controlled conditions (at −5 °C) until the experiment started. In mid-June 2022, the seeds were sown at a rate of three per each pot filled with perlite-peat substrate, placed on the surface of the substrate. Paulownia seeds are very light (weight of 1,000 seeds: ca. 0.17 g) and need light to germinate; thus, the pots were placed in a well-sunlit place in a greenhouse foil tunnel, and water was provided carefully with the aim of not damaging or displacing the seeds on the surface of the substrate. After approximately two weeks, visible germination began. During the period of germination and initial seedling development, pots were watered every day up to field capacity. Excess individuals were removed, leaving one in each pot. Four weeks after the last individuals visually germinated, they were arranged in a randomized block design with irrigation levels as the fixed factor and blocks as the random factor. Then, on August 6th, drought treatment of the seedlings with various levels of soil moisture was initiated. The following soil moisture variants were established: 100%, 75%, 50% and 25% of the field capacity (moisture content of a soil after drainage). Materials were collected three times for biochemical analyses. The first collection took place on August 16th, 10 days after the treatment was initiated, and each subsequent collection was performed in total numbers of three at 10-day intervals (one-day collection taking place every 10 days) up to September 5th. The analyses were performed with leaves from the apical portion of the seedlings (as materials from the aboveground part of the seedling) and lateral roots (as materials from the underground part of the seedling), at each time interval at 10 am. All of the experiments were carried out in triplicate per treatment at each collection date. Proline and $H_2O_2$ levels were determined on the day material was collected. Material obtained for measuring the protein content and enzyme activity was frozen in liquid nitrogen immediately after collection and stored at −80 °C.

### Proline determination

Proline concentrations were determined according to a modified method by *Carillo & Gibon (2011)*. A cold extraction procedure was used to extract proline involving the grinding 20–50 mg fresh weight leaf and root aliquots separately with mortar and subsequently mixing them with 2 ml of ethanol:water (40:60 v/v). The resulting mixture was left overnight at 4 °C and then centrifuged at 14,000× g for 5 min at 4 °C. One milliliter of the reaction mixture containing 1% (w/v) ninhydrin in 60% (v/v) acetic acid and 20% (v/v) ethanol was added to 0.5 ml of the sample aliquots and to the standards and then heated at 95 °C in the block heater for 20 min. The absorbance of the supernatant was read at 520 nm. The proline content of the samples was determined based on a standard curve, according to *Carillo & Gibon (2011)*. L-Proline (P0380; Sigma-Aldrich, Burlington, MA,
USA) solutions of known concentrations diluted in the same medium used for extraction were used as standards.

## H$_2$O$_2$ determination

Levels of H$_2$O$_2$ were determined according to *Alexieva et al. (2001)*. Fresh leaves and roots (20–50 mg) were separately ground with a mortar and mixed with 2 ml of 0.1% trichloroacetic acid (TCA). The homogenate was centrifuged at 12,000× g for 15 min at 4 °C, and then 0.5 ml of the supernatant was added to 0.5 ml of 10 mM potassium phosphate buffer (pH 7.0) and 1 ml of 1 M KI. The absorbance of the supernatant was read at 390 nm. The amount of hydrogen peroxide was calculated using a standard curve prepared with known concentrations of H$_2$O$_2$.

## P5CS activity

The assay of P5CS activity was performed following the method described by *Hayzer & Leisinger (1979)*. Samples (20–50 mg) were homogenized in an extraction buffer (100 mM Tris–HCl, 10 mM MgCl$_2$, 1 mM EDTA, 10 mM $\beta$-mercaptoethanol, 4 mM DTT, 2 mM PMSF and 2% PVPP, pH 7.5). The extracts were centrifuged at 10,000× g for 20 min at 4 °C, and then the supernatant was clarified by centrifugation at 10,000× g for 20 min at 4 °C. The analysis of P5CS activity was carried out in the reaction mixture containing Tris–HCl (50 mM, pH 7.0), 50 mM L-glutamate, 20 mM MgCl$_2$, 100 mM hydroxylamine· HCl, 10 mM ATP and the enzyme extract, incubated at 37 °C for 15 min. The reaction was stopped by adding 1 ml of the stop buffer (2.5 g FeCl$_3$ and 6 g TCA in a final volume of 100 ml of 2.5 N HCl). The mixture was then centrifuged to remove precipitated proteins. The absorbance was read at 535 nm using a spectrophotometer against a blank identical to the one described above but without ATP. Total protein content was determined according to *Bradford (1976)* using Bradford reagent.

## Isolation of mitochondria

To measure the activity of ProDH, which is located in mitochondrial membranes, mitochondria were isolated using methods described by *Małecka et al. (2009)*. The plant material (about 2 g FW) was blended in special buffer (0.35 M sucrose, 1 mM EDTA, 5% BSA, 1% polyvinylpyrrolidone, 0.05 M KH$_2$PO$_4$/K2H$_P$O$_4$ buffer at pH 7.2). The resulting mixture was centrifuged at 3,000× g for 10 min and then the resulting supernatant was clarified by centrifugation at 10,000× g for 20 min. Meanwhile, the pellet was resuspended in a mixture containing 1 mM EDTA, 0.2% BSA, 0.3 M mannitol and 20 mM Mops (3-($N$-morpholino)propanesulfonic acid) at pH 7.2), and subsequently subjected to purification through a gradient using 24% (v/v) Percoll in 0.25 M sucrose, 0.2% BSA and 20 mM Mops (pH 7.2). The gradient was then centrifuged for 30 min at 40,000× g. The fractions of mitochondria and peroxisomes were obtained and washed to remove Percoll in a 20-fold volume of the buffer (0.35 M sucrose, 20 mM MOPS, KH$_2$PO$_4$/K2H$_P$O$_4$ at pH 7.2) and then centrifuged for 30 min at 4,000× g. The purified mitochondria and peroxisomes were resuspended in the same buffer.

### ProDH activity

The activity of ProDH was measured using the 2,6-dichlorophenolindophenol (DCIP)-based assay described by *Huang & Cavalieri (1979)*. The reaction mixture contained 0.05 M Tris–HCl buffer (pH 8.5), 5 mM $MgCl_2$, 0.5 mm FAD, 1 mm KCN, 1 mm phenazine methosulfate, 0.06 mM DCPIP and 0.1 M proline. Mitochondrial preparations with known protein concentrations were added to 1 ml of reaction mixture. The reaction was monitored at 600 nm at 25 °C using proline to initiate the reaction. Total protein content was determined according to *Bradford (1976)* using Bradford reagent.

### Statistical analysis

Data were analyzed using R statistical computing software (*R Core Team, 2022*). Differences between treatments were assessed by means of analysis of variance (ANOVA). Tukey's test was used to determine significant differences between means at $P = 0.05$. Test assumptions were checked using the Shapiro–Wilk test (to assess normal distribution) and Levene's test (to assess the equality of variances). Correlation was assessed using Pearson's method and visualized using the *corrplot* package (*Wei & Simko, 2021*).

## RESULTS

The proline levels in leaves at the 100% moisture treatment was significantly lower from those at the 75% and 50% moisture treatments on the 2nd collection date (Fig. 2A). On other collection dates, the proline content did not show significant changes due to soil moisture. The duration of the experiment affected the proline content in leaves, causing a gradual increase on each subsequent collection date. For the roots, significant differences in proline levels occurred only on the 3rd collection date; the proline content from the 100% moisture treatment was significantly lower than that from the 50% and 25% moisture treatments (Fig. 2A'). The proline content did not show a clear trend of changes during the experiment between the 1st and 2nd collection dates; however, the proline level in each variant decreased between the 2nd and 3rd collection dates. At the overwhelming majority of collection dates, the proline content was not significantly negatively correlated with soil moisture in the leaves and roots (Fig. 3).

The levels of $H_2O_2$ in leaves changed distinctly as the soil moisture decreased on the 1st and 3rd collection dates (Fig. 2B). On the 1st and 3rd dates, the $H_2O_2$ level in the treatment with the highest intensity of drought stress (25% of soil moisture) was significantly higher than in the case of other treatments. The levels of $H_2O_2$ correlated negatively with soil moisture in leaves from each collection date (Figs. 3A, 3B, 3C). During the experiment, the levels of $H_2O_2$ increased, which was noticeable between the 2nd and 3rd collection dates. In roots, the $H_2O_2$ levels increased significantly with decreasing soil moisture on the 1st collection date (Fig. 2B'). There was no noticeable trend of change during the experiment.

The activity of P5CS in leaves increased due to the decreasing soil moisture, as significant differences were noticed at each collection date (especially the 3rd date); at the 25% moisture treatment, the activity was over double that of the 100% moisture treatment (Fig. 2C). In roots, the activity of this enzyme also increased due to the decreasing soil moisture on each collection date, especially between treatments of 100% and 25% soil moisture (Fig. 2C').

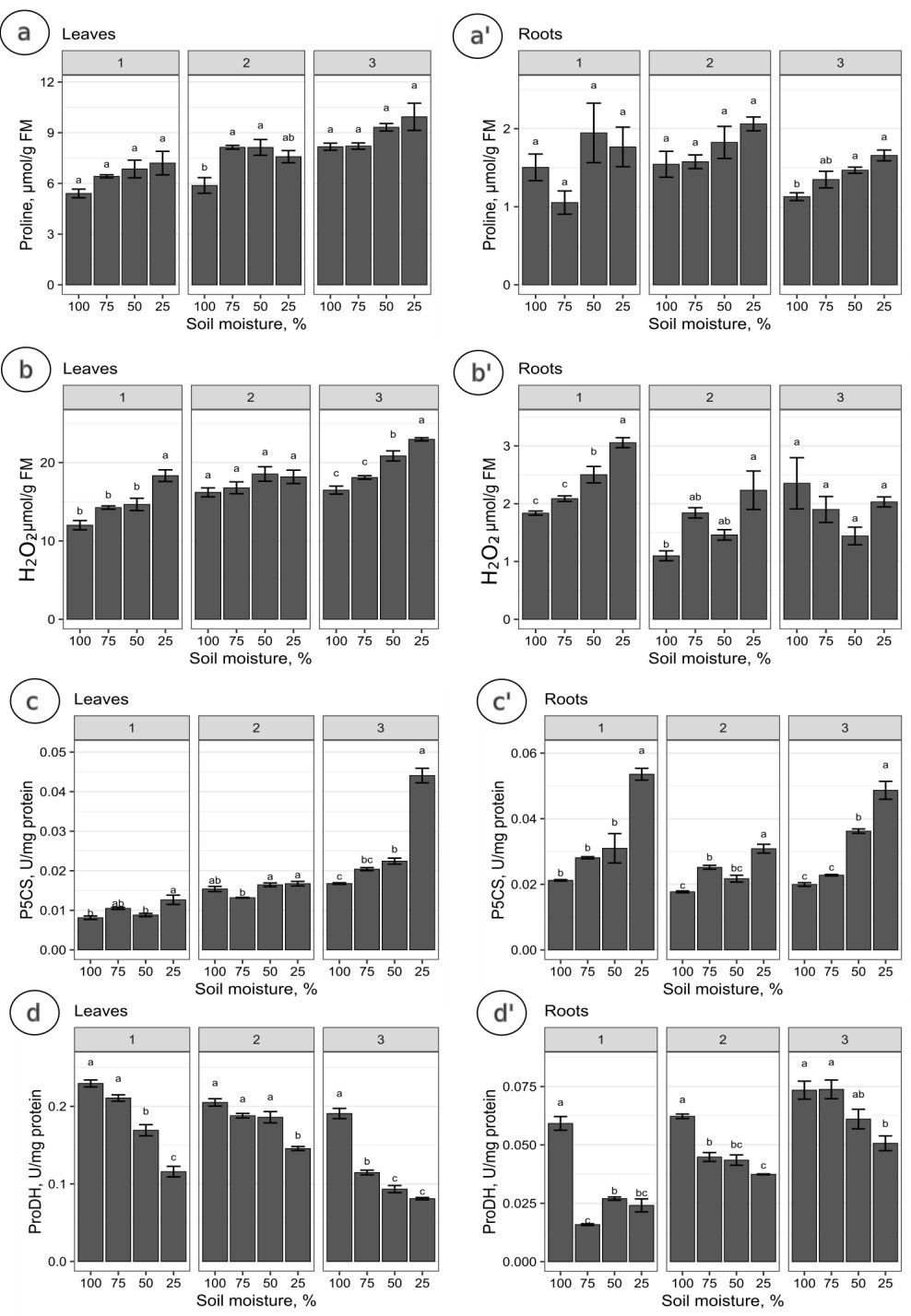

**Figure 2** **The content of proline and the activity of analyzed enzymes in relation to the soil moisture treatments on subsequent collection dates.** The content of proline (A, A'), $H_2O_2$ (B, B'), and the activity of P5CS (C, C') and ProDH (D, D') in relation to the soil moisture treatments on subsequent collection dates (1, 2, 3). A Tukey's test was performed each collection date. Values with different letters are significantly different at $p \leq 0.05$. Mean $\pm$ SE.

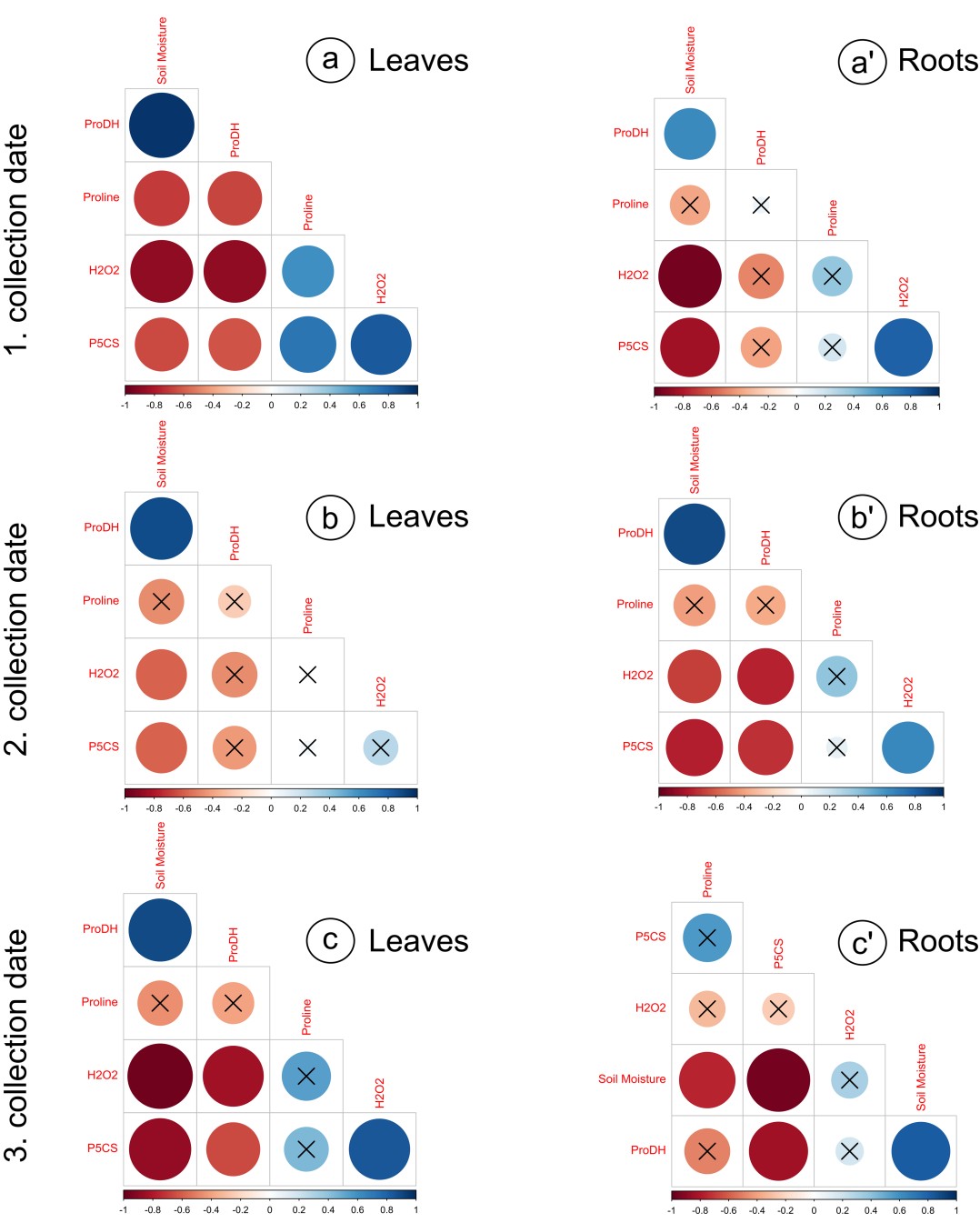

**Figure 3** **Correlation matrices calculated for the analyzed parameters, divided into leaves and roots, on subsequent collection dates.** (A) a': 1. collection date; (B) b': 2. collection date; (C) c': 3. collection date. Crossed numbers indicate a nonsignificant correlation ($p > 0.05$). The more red the color is, the correlation coefficient is closer to $-1$; the more blue color is, the correlation coefficient is closer to 1. Pearson's correlation.

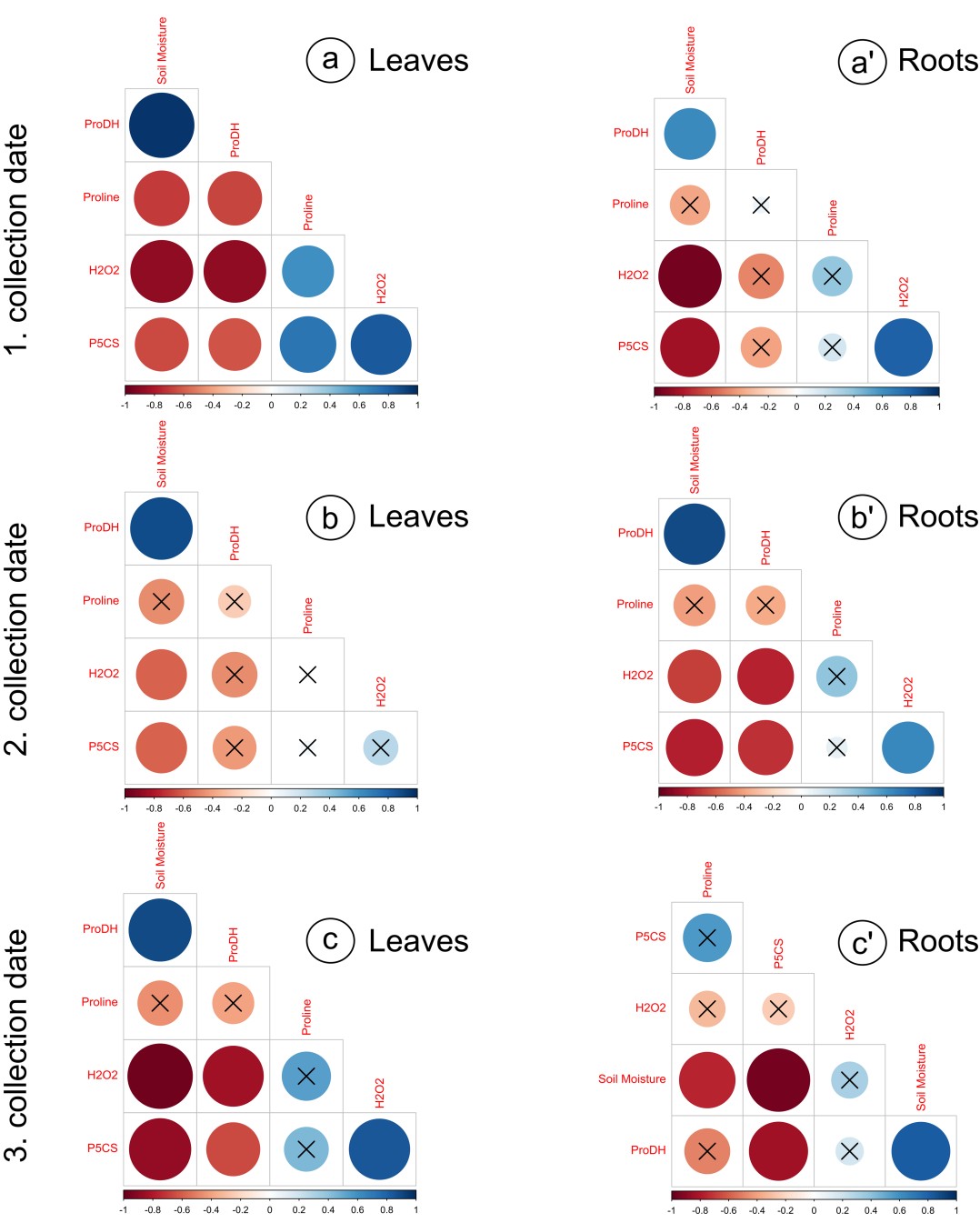

The level of P5CS activity was significantly higher at the 1st date in roots than in leaves—for the 100% moisture treatment, the difference was over double the value, and in the 25% moisture treatment, it was almost five times the value. In both leaves and roots, the activity of P5CS correlated significantly with soil moisture, while the correlations were negative (Fig. 3).

The activity of ProDH in leaves decreased with decreasing soil moisture (Fig. 2D). Differences were significant between extreme moisture treatments (100% and 25% of soil moisture) at each collection date, and for the 1st and 3rd dates, they were also significant between intermediate treatments. In roots, ProDH activity also decreased due to the decrease in soil moisture, especially between treatments with 100% and 25% soil moisture (Fig. 2D'). The level of this enzymatic activity was much higher in the leaves (0.02–0.075 U/mg protein) than in the roots (0.09–0.25 U/mg protein). Significant positive correlations of ProDH activity with soil moisture were observed at each collection date for the leaves and roots (Fig. 3).

## DISCUSSION

Although the biology of plant stress has long been studied in correlation to proline metabolism, how proline influences plant stress and how proline can be used to increase the resistance of plants to drought stress remain unclear (*Szabados & Savoure, 2010*; *Verslues & Sharma, 2010*; *Bhaskara, Yang & Verslues, 2015*). In numerous experiments, it has been assumed that increasing synthesis and decreasing catabolism results in higher levels of proline content and, as a result, increased drought tolerance (*Su & Wu, 2004*; *Tateishi, Nakagawa & Esaka, 2005*; *Molinari et al., 2007*). Although some studies reported that the resistance to stress in herbaceous plant was improved, the levels of water available were not different, and only plants that survived drought stress were counted. In the present study, tree species seedlings and different treatments of soil moisture were used as drought stress tests.

It was observed that the level of drought severity is closely related to the activity of enzymes involved in proline metabolism, but not the content of proline. The activity of P5CS, the major rate-limiting enzyme for proline accumulation, was positively correlated with soil moisture, while the activity of ProDH catalyzing proline catabolism was negatively correlated with soil moisture. This was observed in the leaves and roots at every collection date without exception (Figs. 2C, 2C', 2D, 2D', 3). In contrast, proline content differed significantly due to soil moisture treatments only at the 2nd date in leaves and at the 3rd date in roots, although an upward trend was observed with a decrease in soil moisture in the leaves (but not in the roots) (Figs. 2A, 2A'). Notably, the level of $H_2O_2$ was correlated with soil moisture in leaves and roots from all dates (except for the 3rd collection date in roots) (Fig. 3). This suggests that the seedlings and the soil moisture treatment experienced drought stress at a similar level. As a ROS, $H_2O_2$ plays an essential role in intracellular communication and thus participates in the development of plant adaptation to specific conditions (*El-Maarouf-Bouteau & Bailly, 2008*). Unfavorable environmental conditions, such as heat or drought, disturb cell homeostasis and initiate oxidative stress, which

results in excessive ROS accumulation. This state triggers many protective mechanisms. An example is proline accumulation through the upregulation of P5CS activity, which results from $H_2O_2$ accumulation due to oxidative stress, while ProDH activity decreases (Fig. 1). Thus, it has been shown that $H_2O_2$ is involved in proline metabolism as a regulatory signaling molecule (*Uchida et al., 2002*; *Yang, Lan & Gong, 2009*; *Wen et al., 2013*; *Ben Rejeb, Abdelly & Savouré, 2014*).

It seems that proline content does not directly reflects the degree to which a plant experiences stress or exhibits stress tolerance. Although proline accumulation contributes to stress resistance through several mechanisms, such as scavenging ROS, improving the integrity of cell membranes and preventing water loss, it has been observed that some specific metabolic adjustments were shaped by provenances adapted to drought; as a result, maintaining a high level of proline was no longer necessary (*Kesari et al., 2012*). Thus, adaptation to drier conditions is probably not connected with increased proline accumulation. This also corresponds to the results of our previous studies, in which changes in the level of proline correlated with moisture and thermal conditions only for seeds sensitive to desiccation (*Kijowska-Oberc et al., 2020*). However, as noted by *Bhaskara, Yang & Verslues (2015)*, it remains unclear whether higher or lower proline accumulation correlates with higher or lower flux through proline synthesis and catabolism, and a broader view is needed for proline metabolism as a cycle regulated by multiple cellular mechanisms. Only a few of these cycles are known, such as the ABA-dependent signaling pathway (*Savouré et al., 1997*; *Zdunek-Zastocka et al., 2021*). Although the influence of highly ABA-induced protein phosphatase 2Cs on ProDH and P5CS is different, the proline content is similar (*Sharma & Verslues, 2010*; *Bhaskara, Nguyen & Verslues, 2012*). Moreover, the catabolic pathway is a very important alternative source of energy during stress conditions (*Verslues & Sharp, 1999*). Proline is oxidized by ProDH localized on the matrix side of the mitochondrial inner membrane (*Cabassa-Hourton et al., 2016*). P5CS converts glutamate to proline using NAD(P)H as an energy source, while ProDH participates in conversion back to glutamate using FAD+ and NAD+ (*Liang et al., 2013*). Therefore, proline degradation participates in the regulation of the $NAD(P)^+/NAD(P)H$ ratio in cells *via* the transport of reducing equivalents between the cytoplasm and mitochondria, which is important, especially during plant senescence and prolonged abiotic stress (*Kavi Kishor et al., 2022*). When operating in parallel with proline metabolism, these mechanisms provide an additional influence; as a result, proline metabolism cannot be directly translated into proline content (*Sharma, Villamor & Verslues, 2011*). Therefore, in further studies with different levels of drought stress, the effect of proline metabolism, as well as the mechanisms that regulate proline metabolism, on plant stress tolerance should be different than the effect of proline level.

Some studies have shown that proline biosynthesis due to stress occurs to a greater extent in the roots in meristematic tissue division (*Deuschle et al., 2001*; *Tripathi & Gaur, 2004*; *Ghosh et al., 2022*). Some authors based on their observations suggested that most synthesized proline is transported to the aboveground tissues (*Hua et al., 1997*; *Armengaud et al., 2004*; *Jaiswal et al., 2010*). Higher levels of ProDH activity were found in leaves than in roots under metal stress (*Li et al., 2013*). As confirmed by the present study, this also

occurred for *P. tomentosa* ([Figs. 2D](), [2D'](); [4]()). P5CS activity was overwhelmingly higher in the roots than leaves, which may indicate that these tissues are proline biosynthesis sites, while the activity of ProDH was significantly higher in the leaves than roots. On the other hand, the results of another group of studies suggest that higher proline biosynthesis during stress occurs in leaves; then, proline is exported to the roots and accumulates (*Verslues & Sharp, 1999*; *Ueda, Yamamoto-Yamane & Takabe, 2007*; *Saadia et al., 2012*). It was shown that under conditions of low osmotic potential, proline is synthesized mainly in the photosynthetic tissue, while its catabolism occurs to a greater extent in roots. In meristematic tissues of underground parts of a plant, proline acts as an additional respiratory substrate and supports continued growth under drought conditions, while ProDH serves as a source of energy by proline degradation (*Verslues & Sharp, 1999*). However, the mentioned studies were conducted using herbaceous plants rather than woody plants, and they often involved osmotic and salt stress in addition to drought stress. Moreover, it remains unclear whether proline accumulation is more connected with stress tolerance or maintaining energy and redox homeostasis through the activation of proline catabolism, which enables plant development under long-term stress (*Kavi Kishor & Sreenivasulu, 2014*). *Sharma, Villamor & Verslues (2011)* determined that a model of tissue-specific differences in proline metabolism is fundamental to the protective role of this amino acid in the face of drought. The results of our research show that the level of proline biosynthesis and catabolism are different between leaves and roots of *P. tomentosa* ([Fig. 4]()). The higher ProDH activity in leaves seems to result from the upregulation of this enzyme by high concentrations of proline in leaves (*Misra & Gupta, 2005*). This tissue-specific arrangement may be characteristic of this species (*Hosseinifard et al., 2022*). It has previously been observed, as reported in the present study, that the level of proteins connected with proline metabolism and the proline content correlate with each other in all plant organs except the roots, which was explained by the transport of proline from the roots by the xylem to the shoot (*Hua et al., 1997*; *Jaiswal et al., 2010*). Increased catabolism of proline in leaves may also result from long-term drought stress and initiation of aging, in which the protective role of proline changes into an energetic role (*Launay et al., 2019*). Stress-specific and tissue-specific differences in proline metabolism have already been observed, *e.g.*, the accumulation of proline in unfolded leaf tissue instead of meristematic tissue; these differences may be more important than previously thought (*Ueda, Yamamoto-Yamane & Takabe, 2007*; *Skirycz et al., 2010*; *Sharma, Villamor & Verslues, 2011*). There is an undeniable need to fully characterize proline transport and processes affecting proline levels in parallel with proline metabolism.

## CONCLUSION

For many years, researchers have been searching for a precise marker of osmotic stress-induced damage to plants. This task is difficult because how a plant responds to stress depends on the species, the duration of stress, and many other molecular and biochemical mechanisms. Many studies indicate that proline could function as this marker because the accumulation and metabolism of proline largely determines the plant's response to stress.

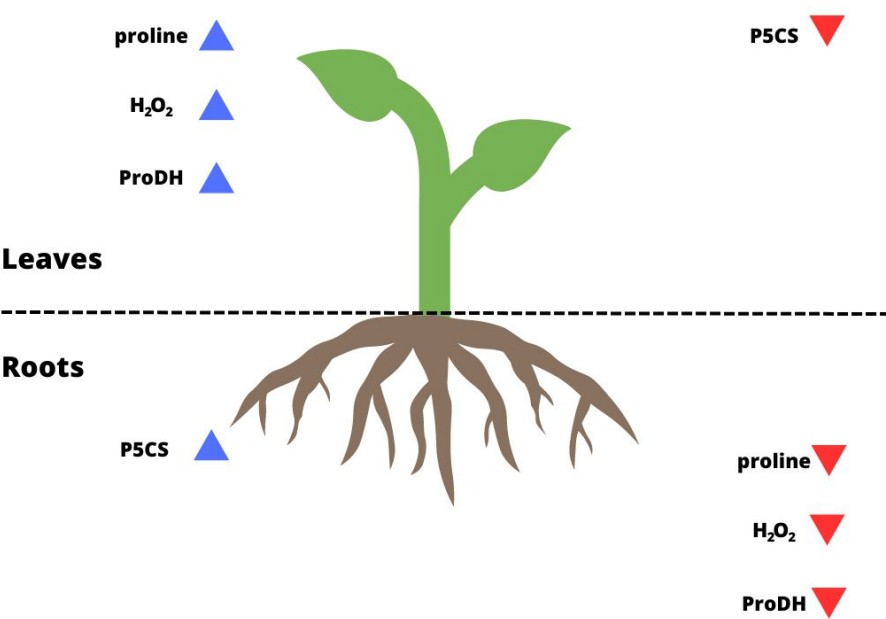

**Figure 4** **Summary of the results for leaves and roots of drought-treated plants.** The blue arrows pointing up indicate parameters that reached a higher level/activity in a given organ; the red arrows pointing down indicate parameters that were at a lower level/activity in a given organ.

Our results showed that the degree of drought stress experienced by *P. tomentosa* was directly affected by the activity of enzymes involved in proline metabolism, but not the content of proline. Moreover, we observed tissue-specific differences in proline metabolism between the aboveground and underground parts of plants, which may be characteristic of *P. tomentosa* or may result from stress and initiation of aging, in which proline plays an energetic role. However, the synthesis and catabolism of proline are influenced by many other elements, including the plant species, activity of a given proline biosynthesis pathway, activity of enzymes responsible for proline biosynthesis and catabolism, proline degradation by ProDH as an additional energy source, influence of ABA-dependent signaling on these enzymes, activity of proline transporters, content of elements in the soil (N, P), and availability of arbuscular mycorrhizae. Although the number of factors is very large, every factor must be considered to conduct analyses related to the role of proline in plant stress tolerance.

We believe that special attention should be focused on the process of proline catabolism in mitochondria and the role of the enzymes responsible for this process, namely, ProDH and pyrroline-5-carboxylate dehydrogenase (P5CDH). Under stress, the main changes occur in the structure of the mitochondrion, and the redox imbalance can interfere with proline catabolism. On the other hand, proline degradation in mitochondria by ProDH provides an alternative source of energy during cell division or senescence. In addition to proline catabolism, P5CDH produces NADPH, which may be used by thioredoxin (Trx) proteins to maintain redox balance. Moreover, Trx affects the activity of proline cycle

enzymes. Properly functioning redox systems guarantee that metabolic processes correctly occur in plant cells. As a result, plants adapts more quickly to stress conditions, which is of particular importance with the limited water availability resulting from global warming.

## ACKNOWLEDGEMENTS

The authors would like to thank Danuta Ratajczak, Agata Obarska and Paulina Pilarz for providing technical assistance. We would also like to thank Prof. Aneta Piechalak for providing substantive assistance and Prof. Paweł Chmielarz for helping with establishment of the experiment.

### Funding

This research was supported by the Institute of Dendrology, Polish Academy of Sciences (2022/02/ZB/FBW/00001). The funders played a role in data collection and decision to publish. The funders had no role in study design, data analysis, or preparation of the manuscript.

### Grant Disclosures

The following grant information was disclosed by the authors:
The Institute of Dendrology, Polish Academy of Sciences: 2022/02/ZB/FBW/00001.

### Competing Interests

The authors declare there are no competing interests.

### Author Contributions

- Joanna Kijowska-Oberc conceived and designed the experiments, performed the experiments, analyzed the data, prepared figures and/or tables, authored or reviewed drafts of the article, and approved the final draft.
- Mikołaj K. Wawrzyniak conceived and designed the experiments, analyzed the data, prepared figures and/or tables, authored or reviewed drafts of the article, and approved the final draft.
- Liliana Ciszewska performed the experiments, authored or reviewed drafts of the article, and approved the final draft.
- Ewelina Ratajczak conceived and designed the experiments, analyzed the data, authored or reviewed drafts of the article, and approved the final draft.

### Data Availability

  The raw measurements are available in the Supplementary File.

### Supplemental Information

Supplemental information for this article can be found online at http://dx.doi.org/10.7717/peerj.16697#supplemental-information.

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
