# Peer review of "Evaluation of P5CS and ProDH activity in Paulownia tomentosa (Steud.) as an indicator of oxidative changes induced by drought stress"

_PeerJ, doi:10.7717/peerj.16697_

## Round 0.1 · original submission · Minor Revisions

Both reviewers did not raise any issue regarding experimental design; most issues refer to clarify some doubts in the text.

Reviewer 1 ·

Basic reporting

In the abstract, full sentence should be used without omitting "proline", even there is context.
"Moreover, we noted tissue-specificc differences in this species, in which roots appeared to be biosynthesis sites and leaves appeared to be catabolism sites."

Contradictory description is found from line 181 to line 184 about proline levels in leaves.
"181 The proline levels in leaves changed insignificantly due to soil moisture [Fig. 2a]. Although it
182 increased with decreasing soil moisture, the proline content at the 100% moisture treatment was
183 significantly lower from those at the 75% and 50% moisture treatments only on the 2nd collection
184 date."

Experimental design

No comment.

Validity of the findings

Authors described root proline level as decreased in 3rd collection date compared to 2nd collection date, but no statistical analysis was performed between these two collection dates. I only see the ANOVA-Tukey test performed within each collection date. If statistical analysis was not performed between collection dates, then authors should not say the proline level decreases in general in roots, both for figure 2a’ and figure 5.

No data support the proline transport hypothesis in the discussion of line 321. It could be that they are both sites of synthesis and the leaves synthesize more. The general reading of concentration of leaf proline is higher in leaves than in roots, and the precursor of proline biosynthesis could come from leaf chloroplast. Stating the root to be the site of proline biosynthesis site is really confusing and lack of experimental evidence.
"321 Moreover, we observed clear tissue-specific differences in proline metabolism between
322 the aboveground and underground parts of plants, which suggests that proline may be transported
323 from the roots to shoots of P. tomentosa seedlings."

·

Basic reporting

The authors are interested in investigating the levels of proline and activities of enzymes associated with proline metabolism in the leaves and roots of Paulownia tomentosa tree seedlings under drought conditions, especially at different time periods post drought treatment. Paulownia tomentosa has important commercial value and with the increasing water shortage, it is important to understand the physiology of drought sensitive tree seedlings. The authors have taken a good approach to their study and their findings demonstrate that monitoring the activity of the enzymes associated with proline biosynthesis can provide valuable information than just the proline content itself. The authors have provided sufficient background information, shared raw data, and the manuscript is well written. However, there is some disconnect in the method section, especially the way drought treatment was provided. The authors need to re-work on explaining their methodology and probably provide figures or pictures to minimize the confusion.

Experimental design

The research in the manuscript meets with the aims and scope of the journal. The method is described well but it needs improvement, especially with the explanation of drought treatment. Below are my comments/ questions/ suggestions that will be helpful in improving the content of the method section-

Experimental design
Line 97: How deep were the seeds sown?
Line 100: What was the day/night duration, temperature, humidity?
Line 101: How many seeds were there in a pot?
Line 104 – 105: The drought treatment was started. Following irrigation levels were established: 100%.....25%. How can 100% irrigation be drought? The term irrigation level is confusing. Wouldn’t 75 -25% relative soil moisture/water content?
Line 106-107: The first collection took place 10 days after drought was initiated…..again it is confusing to consider 100% irrigation as drought? The collection was done at 10 days interval for how long? One month? Was the collection of the sample at each time interval done at the same time? Needs to be specified. Probably showing a time line and depicting how long it took for seeds to germinate, when the drought was initiated and when the sampling was done, etc. will be helpful.
Line 106: It would be helpful if the authors could provide a brief definition of field capacity.
Line 108: The analyses were performed with fully expanded leaves…. Were the leaves fully expanded in drought treated seedlings? This statement is confusing. The leaves were not shriveled in drought stressed seedlings? What is the significance of 10 day intervals? Are there three 10-day interval collections (mentioned as collection 1, 2 3). Needs to be properly explained/ associated.

Proline Determination
Line 118: Was the leaf and root tissue powdered or cut in small pieces?
Line 124-125: What type of proline was used as standard? Company? What was the range of the concentration of the standard? Please provide formula for determining the amount of proline.

H2O2 Determination
Line 132: What type of H2O2 was used as standard? Concentration of H2O2? What was the range of concentration for standard plot. What is the formula for calculation of H2O2?

P5CS activity
Line 135: How much sample was homogenized?
Isolation of mitochondria

Line 149: How much material? Leaf and roots?

Validity of the findings

The data is robust and the discussion and conclusion is well stated. I have few questions from Figure 2 that probably needs to be addressed, perhaps in the discussion...

Figure - 2: What could be possible explanation for proline content in 100% irrigated leaves of 3rd collection higher than the first and second collection? Isn’t the 100% irrigated sample common for all collections time points? Similar changes observed in H2O2 content in roots.

Line 194: drought stress (25% of soil moisture) was significantly different from that…..What was the difference? Higher or lower, needs to be mentioned. Similarly for line 195.

Additional comments

Below are a few of my comments/questions/suggestions for abstract and introduction sections-

Abstract:
Instead of ‘Developed’ Paulownia tomentosa seedlings…., It will be helpful to mention the age of the seedling that was used for the study.
Developed Paulownia tomentosa seedlings were exposed to drought conditions at various levels (irrigation at 100, 75, 50, and 25% of field capacity)…..This statement is confusing. Is 100% irrigation drought condition?
The material was collected at ten days interval…..It will be helpful to state here for how long the plant material was collected following drought treatment.
It will be helpful to mention which plant tissues were assayed for proline content and enzyme activities.

Introduction:
Line 38-39: I think it is important to mention here that proline accumulates at high concentrations in cells and prevents loss of proper turgor during water stress by making osmotic adjustments.

---

## Round 0.2 · accepted · Accept

The contributors have effectively responded to all feedback from the referees. The paper is suitable for publication.

Reviewer 1 ·

Basic reporting

No comment.

Experimental design

No comment.

Validity of the findings

No comment.

Additional comments

Authors fixed the questions raised by both reviewers. I think the manuscript is in good shape for publication.

·

Basic reporting

No comment.

Experimental design

No comment.

Validity of the findings

No comment.

Additional comments

I like to thank the authors for revising their manuscript and including the requested information. Their manuscript is well written and informative.